# CUBIC SPLINE SMOOTHING COMPENSATION FOR IRREGULARLY SAMPLED SEQUENCES

## ABSTRACT

The marriage of recurrent neural networks and neural ordinary differential networks (ODE-RNN) is effective in modeling irregularly sampled sequences. While ODE produces the smooth hidden states between observation intervals, the RNN will trigger a hidden state jump when a new observation arrives and thus cause the interpolation discontinuity problem. To address this issue, we propose the *cubic spline smoothing compensation*, which is a stand-alone module upon either the output or the hidden state of ODE-RNN and can be trained end-to-end. We derive its analytical solution and provide its theoretical interpolation error bound. Extensive experiments indicate its merits over both ODE-RNN and cubic spline interpolation.

## 1 INTRODUCTION

Recurrent neural networks (RNNs) are commonly used for modeling regularly sampled sequences (Cho et al., 2014). However, the standard RNN can only process discrete series without considering the unequal temporal intervals between sample points, making it fail to model irregularly sampled time series commonly seen in domains, e.g., healthcare (Rajkomar et al., 2018) and finance (Fagereng & Halvorsen, 2017). While some works adapt RNNs to handle such irregular scenarios, they often assume an exponential decay (either at the output or the hidden state) during the time interval between observations (Che et al., 2018; Cao et al., 2018), which may not always hold.

To remove the exponential decay assumption and better model the underlying dynamics, Chen et al. (2018) proposed to use the neural ordinary differential equation (ODE) to model the continuous dynamics of hidden states during the observation intervals. Leveraging a learnable ODE parametrized by a neural network, their method renders higher modeling capability and flexibility.

However, an ODE determines the trajectory by its initial state, and it fails to adjust the trajectory according to subsequent observations. A popular way to leverage the subsequent observations is ODE-RNN (Rubanova et al., 2019; De Brouwer et al., 2019), which updates the hidden state upon observations using an RNN, and evolves the hidden state using an ODE between observation intervals. While ODE produces smooth hidden states between observation intervals, the RNN will trigger a hidden state jump at the observation point. This inconsistency (discontinuity) is hard to reconcile, thus jeopardizing continuous time series modeling, especially for interpolation tasks (Fig. 1 top-left).

We propose a *Cubic Spline Smoothing Compensation (CSSC)* module to tackle the challenging discontinuity problem, and it is especially suitable for continuous time series interpolation. Our CSSC employs the cubic spline as a means of compensation for the ODE-RNN to eliminate the jump, as illustrated in Fig. 1 top-right. While the latent ODE (Rubanova et al., 2019) with an encoder-decoder structure can also produce continuous interpolation, CSSC can further ensure the interpolated curve pass strictly through the observation points. Importantly, we can derive the closed-form solution for CSSC and obtain its interpolation error bound. The error bound suggests two key factors for a good interpolation: the time interval between observations and the performance of ODE-RNN. Furthermore, we propose the *hidden CSSC* that aims to compensate for the hidden state of ODE-RNN (Fig. 1 bottom), which not only assuage the discontinuity problem but is more efficient when the observations are high-dimensional and only have continuity on the semantic level. We conduct extensive experiments and ablation studies to demonstrate the effectiveness of CSSC and hidden CSSC, and both of them outperform other comparison methods.

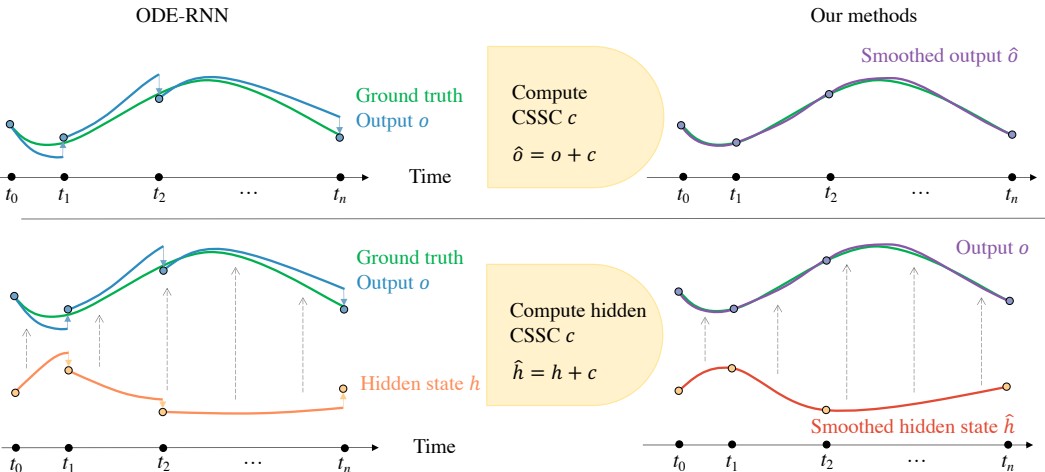

Figure 1: The illustration of ODE-RNN and our methods. The top left is ODE-RNN showing the interpolation curve jump at the observation points. The top right is the smoothed output by our CSSC, where the jump is eliminated, and the output strictly passes the observation. The bottom left shows the ODE-RNN 's discontinuous output caused by the hidden state discontinuity. The bottom right shows that our hidden CSSC is applied to the hidden state of ODE-RNN, resulting in the smooth hidden state and so as the output.

## 2 RELATED WORK

Spline interpolation is a practical way to construct smooth curves between a number of points (De Boor et al., 1978), even for unequally spaced points. Cubic spline interpolation leverages the piecewise third order polynomials to avoid the Runge's phenomenon (Runge, 1901) and is applied as a classical way to impute missing data (Che et al., 2018).

Recent literature focuses on adapting RNNs to model the irregularly sampled time series, given their strong modeling ability. Since standard RNNs can only process discrete series without considering the unequal temporal intervals between sample points, different improvements were proposed. One solution is to augment the input with the observation mask or concatenate it with the time lag $\Delta t$ and expect the network to use interval information $\Delta t$ in an unconstrained manner (Lipton et al., 2016; Mozer et al., 2017). While such a flexible structure can achieve good performance under some circumstances (Mozer et al., 2017), a more popular way is to use prior knowledge for missing data imputation. GRU-D (Che et al., 2018) imputes missing values with the weighted sum of exponential decay of the previous observation and the empirical mean. Shukla & Marlin (2019) employs the radial basis function kernel to construct an interpolation network. Cao et al. (2018) let hidden state exponentially decay for non-observed time points and use bi-directional RNN for temporal modeling.

Another track is the probabilistic generative model. Due to the ability to model the missing data's uncertainty, Gaussian processes (GPs) are adopted for missing data imputing (Futoma et al., 2017; Tan et al., 2020; Moor et al., 2019). However, this approach introduced several hyperparameters, such as the covariance function, making it hard to fine-tune in practice. Neural processes (Garnelo et al., 2018) eliminate such constraints by introducing a global latent variable that represents the whole process. Generative adversarial networks are also adopted for imputing (Luo et al., 2018).

Recently, neural ODEs (Chen et al., 2018) utilize a continuous state transfer function parameterized by a neural network to learn the temporal dynamics. Rubanova et al. (2019) combine the RNN and ODE to reconcile both the new observation and latent state evolution between observations. De Brouwer et al. (2019) update the ODE with the GRU structure with Bayesian inference at observations.

While the ODE produces the smooth hidden states between observation intervals, the RNN will trigger a jump of the hidden state at the observation point, leading to a discontinuous hidden state along the trajectory. This inconsistency (discontinuity) is hard to reconcile, thus jeopardizing the modeling of continuous time series, especially for interpolation tasks. The neural CDE (Kidger

et al., 2020) directly apply cubic splines interpolation at the input sequence to make the sparse input continuous and thus produce continuous output. On the contrary, our method tackles this jumping problem by introducing the cubic spline as a compensation for the vanilla ODE-RNN, at either the output or hidden space.

## 3 METHODS

In this section, we first formalize the irregularly sampled time series interpolation problem (Sec. 3.1), then introduce the background of ODE-RNN (Sec. 3.2). Based upon ODE-RNN, we present CSSC and its closed-form solution (Sec. 3.3), and illustrate the inference and training procedure (Sec. 3.4). Finally, we provide the interpolation error bound of CSSC (Sec. 3.5) and describe an useful extension of CSSC (Sec. 3.6).

### 3.1 PROBLEM DEFINITION

We focus on the interpolation task. Given an unknown underlying function $\mathbf{x}(t) : \mathbb{R} \to \mathbb{R}^d$, $t \in [a, b]$, and a set of $n + 1$ observations $\{\mathbf{x}_k | \mathbf{x}_k = \mathbf{x}(t_k)\}_{k=0}^n \in \mathbb{R}^d$ sampled from $\mathbf{x}(t)$ at the irregularly spaced time points $\Pi : a = t_0 < t_1 < ... < t_n = b$, the goal is to learn a function $F(t) : \mathbb{R} \to \mathbb{R}^d$ to approximate $\mathbf{x}$, such that $F(t_k) = \mathbf{x}_k$.

### 3.2 BACKGROUND OF ODE-RNN

ODE-RNN (Rubanova et al., 2019) achieves the interpolation by applying ODE and RNN interchangeably through a time series, illustrated in top-left of Fig. 1. The function $F$ on time interval $t \in [t_k, t_{k+1})$ is described by a neural ODE with the initial hidden state $\mathbf{h}(t_k)$:

$$\dot{\mathbf{h}}(t) = f(\mathbf{h}(t)); \tag{1}$$
$$\mathbf{o}(t) = g(\mathbf{h}(t)), \tag{2}$$

where the $\mathbf{h} \in \mathbb{R}^m$ is the hidden embedding of the data, $\dot{\mathbf{h}} = \frac{d\mathbf{h}}{dt}$ is the temporal derivative of the hidden state, $\mathbf{o} \in \mathbb{R}^d$ is the interpolation output of $F(t)$. Here, $f : \mathbb{R}^m \to \mathbb{R}^m$ and $g : \mathbb{R}^m \to \mathbb{R}^d$ are the transfer function and the output function parameterized by two neural networks, respectively. At the observation time $t = t_k$, the hidden state will be updated by an RNN as:

$$\mathbf{h}(t_k) = \text{RNNCell}(\mathbf{h}(t_k^-), \mathbf{x}_k); \tag{3}$$
$$\mathbf{o}(t_k) = g(\mathbf{h}(t_k)), \tag{4}$$

where the input $\mathbf{x} \in \mathbb{R}^d$, $t_k^-$ and $t_k^+$ are the left- and right-hand limits of $t_k$. The above formulation has two downsides. The first is the discontinuity problem: while the function described by ODE is right continuous $\mathbf{o}(t_k) = \mathbf{o}(t_k^+)$, the RNN cell in Eq. (3) renders the hidden state discontinuity $\mathbf{h}(t_k^-) \neq \mathbf{h}(t_k^+)$ and therefore output discontinuity $\mathbf{o}(t_k^-) \neq \mathbf{o}(t_k^+)$. The second is that the model cannot guarantee $\mathbf{o}(t_k) = \mathbf{x}_k$ without explicit constraints.

### 3.3 CUBIC SPLINE SMOOTHING COMPENSATION

To remedy the two downsides, we propose the module *Cubic Spline Smoothing Compensation (CSSC)*, manifested in the top-right of Fig. 1. It computes a compensated output $\hat{\mathbf{o}}(t)$ as:

$$\hat{\mathbf{o}}(t) = \mathbf{c}(t) + \mathbf{o}(t), \tag{5}$$

where $\mathbf{o}(t)$ is the ODE-RNN output, and the $\mathbf{c}(t)$ is a compensation composed of piecewise continuous functions. Our key insight is that adding another continuous function to the already piecewise continuous $\mathbf{o}(t)$ will ensure the global continuity. For simplicity, we set $\mathbf{c}(t)$ as a piecewise polynomials function and then narrow it to a piecewise cubic function since it is the most commonly used polynomials for interpolation (Burden & Faires, 1997). As the cubic spline is computed for each dimension of $\mathbf{c}$ individually, w.l.o.g., we will discuss one dimension of the $\mathbf{o}, \mathbf{c}, \hat{\mathbf{o}}, \mathbf{x}$ and thus denote them as $o, c, \hat{o}, x$, respectively. $c(t)$ is composed with pieces as $c(t) = \sum_{k=0}^{n-1} c_k(t)$ with each piece $c_k$ defined at domain $[t_k, t_{k+1}]$. To guarantee the smoothness, we propose four constraints to $\hat{o}(t)$:

1. $\hat{o}(t_k^-) = \hat{o}(t_k^+) = x_k$, $k = 1, ..., n-1$, $\hat{o}(t_0) = x_0$, $\hat{o}(t_n) = x_n$ (output continuity);

2. $\dot{\hat{o}}(t_k^-) = \dot{\hat{o}}(t_k^+)$, $k = 1, ..., n-1$ (first order output continuity);

3. $\ddot{\hat{o}}(t_k^-) = \ddot{\hat{o}}(t_k^+)$, $k = 1, ..., n-1$ (second order output continuity);

4. $\ddot{\hat{o}}(t_0) = \ddot{\hat{o}}(t_n) = 0$ (natural boundary condition).

The constraint 1 ensures the interpolation curves continuously pass through the observations. Constraint 2 and 3 enforce the first and second-order continuity at the observation points, which usually holds when the underline curve $x$ is smooth. And constraint 4 specifies the natural boundary condition owing to the lack of information of the endpoints (Burden & Faires, 1997).

Given $o(t)$ and such four constraints, $c(t)$ has unique analytical solution expressed in Theorem 1.

**Theorem 1.** *Given the first order and second order jump difference of ODE-RNN as*

$$\dot{r}_k = \dot{o}(t_k^+) - \dot{o}(t_k^-); \tag{6}$$

$$\ddot{r}_k = \ddot{o}(t_k^+) - \ddot{o}(t_k^-). \tag{7}$$

*where the analytical expression of $\dot{o}$ and $\ddot{o}$ can be obtained as*

$$\dot{o} = \frac{\partial g}{\partial \mathbf{h}} f; \quad \ddot{o} = f^\mathsf{T} \frac{\partial^2 g}{\partial \mathbf{h}^2} f + \frac{\partial g}{\partial \mathbf{h}}^\mathsf{T} \frac{\partial f}{\partial \mathbf{h}} f, \tag{8}$$

*and the error defined as*

$$\epsilon_k^+ = x_k - o(t_k^+); \tag{9}$$

$$\epsilon_k^- = x_k - o(t_k^-), \tag{10}$$

*then $c_k$ can be uniquely determined as*

$$c_k(t) = \frac{M_{k+1} + \ddot{r}_{k+1} - M_k}{6\tau_k}(t - t_k)^3 + \frac{M_k}{2}(t - t_k)^2 +$$

$$(\frac{\epsilon_{k+1}^- - \epsilon_k^+}{\tau_k} - \frac{\tau_k(M_{k+1} + \ddot{r}_{k+1} + 2M_k)}{6})(t - t_k) + \epsilon_k^+, \tag{11}$$

*where $M_k$ is obtained as*

$$\mathbf{M} = A^{-1}\mathbf{d}, \tag{12}$$

$$A = \begin{pmatrix} 2 & \lambda_1 & & & \\ \mu_2 & 2 & \lambda_2 & & \\ & \ddots & \ddots & \ddots & \\ & & \mu_{n-2} & 2 & \lambda_{n-2} \\ & & & \mu_{n-1} & 2 \end{pmatrix}, \mathbf{M} = \begin{pmatrix} M_1 \\ M_2 \\ \vdots \\ M_{n-2} \\ M_{n-1} \end{pmatrix}, \mathbf{d} = \begin{pmatrix} d_1 \\ d_2 \\ \vdots \\ d_{n-2} \\ d_{n-1} \end{pmatrix}, \tag{13}$$

$\tau_k = t_{k+1} - t_k$, $\mu_k = \frac{\tau_{k-1}}{\tau_{k-1} + \tau_k}$, $\lambda_k = \frac{\tau_k}{\tau_{k-1} + \tau_k}$, $d_k = 6\frac{\epsilon[t_k^+, t_{k+1}^-] - \epsilon[t_{k-1}^+, t_k^-]}{\tau_{k-1} + \tau_k} + \frac{6\dot{r}_k - 2\ddot{r}_k\tau_{k-1} - \ddot{r}_{k+1}\tau_k}{\tau_{k-1} + \tau_k}$, $\epsilon[t_k^+, t_{k+1}^-] = \frac{\epsilon_{k+1}^- - \epsilon_k^+}{\tau_k}$, $M_0 = M_n = 0$.

The proof for Theorem. 1 is in Appx. A. The $\mathbf{c}(t)$ is obtained by computing each $c(t)$ individually according to Theorem. 1.

**Computational Complexity**. The major cost is the inverse of $A$, a tridiagonal matrix, whose inverse can be efficiently computed in $O(n)$ complexity with the tridiagonal matrix algorithm (implementation detailed in Appx. C.1). Another concern is that $\dot{o}$ and $\ddot{o}$ needs to compute Jacobian and Hessian in Eq. (8). We can circumvent this computing cost by computing the numerical derivative or an empirical substitution, detailed in Appx. C.2.

**Model Reduction**. Our CSSC can reduce to cubic spline interpolation if setting $\mathbf{o}$ in Eq. (5) as zero. In light of this, we further analyze our model with techniques used for cubic spline interpolation and experimentally show our advantages against it in Sec. 4.2.

### 3.4 Inference and Training

For inference, firstly compute the predicted value $\mathbf{o}$ from ODE-RNN (Eq. (1-4)), then calculate the compensation $\mathbf{c}$ with CSSC(Eq. (11)); thus yielding smoothed output $\hat{\mathbf{o}}$ (Eq. (5)). For training, the CSSC is a standalone nonparametric module (since we have its analytical solution) on top of the ODE-RNN that allows the end-to-end training for ODE-RNN parameters. We employ Mean Squire Error (MSE) loss to supervise $\hat{\mathbf{o}}$. In addition, we expect the compensation $\mathbf{c}$ to be small to push ODE-RNN to the leading role for interpolation and take full advantage of its model capacity. Therefore a 2-norm penalty for $\mathbf{c}$ is added to construct the final loss:

$$\mathcal{L} = \frac{1}{N} \sum_{i=1}^{N} (||\mathbf{x}(t_i) - \hat{\mathbf{o}}(t_i)||^2 + \alpha ||\mathbf{c}(t_i)||^2) \tag{14}$$

The ablation study (Sec. 4.5) shows that the balance weight $\alpha$ can effectively arrange the contribution of ODE-RNN and CSSC.

**Gradient flow**. Although $\mathbf{c}$ is non-parametric module, but the gradient can flow from it into the ODE-RNN because $\mathbf{c}(t)$ depends on the left and right limit of $\mathbf{o}(t_k), \dot{\mathbf{o}}(t_k), \ddot{\mathbf{o}}(t_k)$. We further analyze that $\dot{\mathbf{o}}(t_k)$ plays a more important role than $\ddot{\mathbf{o}}(t_k)$ in the contribution to $\mathbf{c}(t)$, elaborated in Appx. C.3.

### 3.5 Interpolation Error Bound

With CSSC, we can even derive an interpolation error bound, which is hard to obtain for ODE-RNN. Without loss of generality, we analyze one dimension of $\hat{\mathbf{o}}$, which is scalable to all dimensions.

**Theorem 2.** *Given the CSSC at the output space as Eq. (5), if $x \in C^4[a, b]$, $f \in C^3$, $g \in C^4$, then the error and the first order error are bounded as*

$$||(x - \hat{o})^{(r)}||_\infty \leq C_r ||(x - o)^{(4)}||_\infty \tau^{4-r}, \quad (r = 0, 1), \tag{15}$$

*where $|| \cdot ||_\infty$ is uniform norm, $(\cdot)^r$ is r-th derivative, $C_0 = \frac{5}{384}$, $C_1 = \frac{1}{24}$, $\tau$ is the maximum interval over $\Pi$.*

The proof of Theorem 2 is in Appx. B. The error bound guarantee the error can converge to zero if $\tau \to 0$ or $||(x - o)^{(4)}|| \to 0$. This suggests that a better interpolation can come from a denser observation or a better ODE-RNN output. Interestingly, Eq. (15) can reduce to the error bound for cubic spline interpolation (Hall & Meyer, 1976) if $o$ is set zero. Compared with ODE-RNN, which lacks the convergence guarantee for $\tau$, our model more effectively mitigates the error for the densely sampled curve at complexity $O(\tau^4)$ ; compared with the cubic spline interpolation, our error bound has an adjustable $o$ that can leads smaller $||(x - o)^{(4)}||$ than $||x^{(4)}||$.

An implicit assumption for this error bound is that the $x$ should be 4-th order derivable; hence this model is not suitable for sharply changing signals.

### 3.6 Extend to Interpolate Hidden State

Although CSSC has theoretical advantages from the error bound, it is still confronted with two challenges. Consider the example of video frames: each frame is high-dimensional data, and each pixel is not continuous through the time, but the spatial movement of the content (semantic manifold) is continuous. So the first challenge is that CSSC has linear complexity w.r.t. data dimension, which is still computation demanding when the data dimension becomes very high. The second challenge is that CSSC assumes the underlying function $\mathbf{x}$ is continuous, and it cannot handle the discontinuous data that is continuous in its semantic manifold (e.g. video).

To further tackle these two challenges, we propose a variant of CSSC that is applied to the hidden states, named as *hidden CSSC*, illustrated in bottom of Fig. 1. As we only compute the compensation to hidden state, which can keep a fixed dimension regardless the how large the data dimension is, the computational complexity of hidden CSSC is weakly related to data dimension. Also, the hidden state typically encodes a meaningful low-dimensional manifold of the data. Hence smoothing the hidden state is equivalent to smoothing the semantic manifold to a certain degree.

Hence, the Eq. (5) is modified to $\hat{\mathbf{h}}(t) = \mathbf{c}(t) + \mathbf{h}(t)$, and the output (Eq. (2),(4)) to $\mathbf{o}(t) = g(\hat{\mathbf{h}}(t))$ However, since there is no groundtruth for the hidden state like the constraint 1, we assume the $\mathbf{h}(t_k^+)$

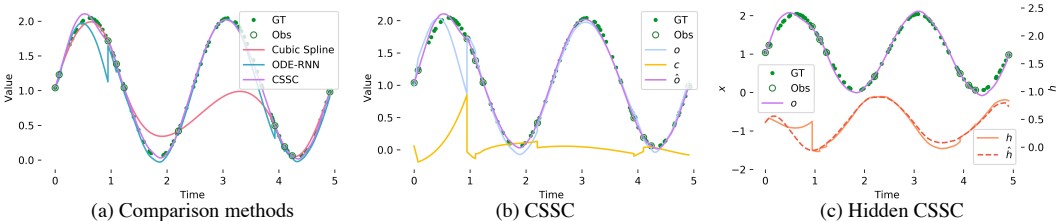

Figure 2: The visual result for Sinuous Wave dataset with 10 observations. (a) shows the comparison of CSSC against other methods; (b) demonstrates the effect of $c$ to smooth the $o$; (c) shows the hidden CSSC output is smooth because its unsmooth hidden state $h$ is smoothed into $\hat{h}$.

are the knots passed through by the compensated curve, rendering the constraints 1 as $\hat{\mathbf{h}}(t_k^-) = \hat{\mathbf{h}}(t_k^+) = \mathbf{h}(t_k^+)$. The rationality for the knots is that $\mathbf{h}(t_k^+)$ is updated by $\mathbf{x}(t_k)$ and thus contain more information than $\mathbf{h}(t_k^-)$. Given the above redefined variable, the closed-form solution of $\mathbf{c}$ is provided in Theorem. 3.

**Theorem 3.** *Given the second order jump at hidden state as*

$$\dot{\mathbf{r}}_k = \dot{\mathbf{h}}(t_k^+) - \dot{\mathbf{h}}(t_k^-); \qquad \ddot{\mathbf{r}}_k = \ddot{\mathbf{h}}(t_k^+) - \ddot{\mathbf{h}}(t_k^-), \tag{16}$$

*where* $\dot{\mathbf{h}} = f$, $\ddot{\mathbf{h}} = \frac{\partial f}{\partial \mathbf{h}} f$, *and the error defined as*

$$\boldsymbol{\epsilon}_k^+ = \mathbf{h}(t_k^+) - \mathbf{h}(t_k^+) = \mathbf{0}; \qquad \boldsymbol{\epsilon}_k^- = \mathbf{h}(t_k^+) - \mathbf{h}(t_k^-), \tag{17}$$

*the* $c_k$ *is uniquely determined as Eq. (11).*

Theorem 3 suggests another prominent advantage: hidden CSSC can be more efficiently implemented because its computation does not involve Hessian matrix.

## 4 EXPERIMENTS

### 4.1 BASELINE

We compare our method with baselines: (1) Cubic Spline Interpolation (Cubic Spline) (2) A classic RNN where $\Delta t$ is concatenated to the input (RNN-$\Delta t$) (3) GRU-D (Che et al., 2018) (4) ODE-RNN (Rubanova et al., 2019) which is what our compensation adds upon (5) Latent-ODE (Rubanova et al., 2019) which is a VAE structure employing ODE-RNN as the encoder and ODE as decoder (6) GRU-ODE-Bayes (De Brouwer et al., 2019) which extends the ODE as a continuous GRU. Our proposed methods are denoted as CSSC for data space compensation and hidden CSSC for hidden space compensation. Implementation Details is in Appx. D.

### 4.2 TOY SINUOUS WAVE DATASET

The toy dataset is composed of 1,000 periodic trajectories with variant frequency and amplitude. Following the setting of Rubanova et al. (2019), each trajectory contains 100 irregularly-sampled time points with the initial point sampled from a standard Gaussian distribution. Fixed percentages of observations are randomly selected with the first and last time points included. The goal is to interpolate the full set of 100 points.

The interpolation error on testing data is shown in Table 1, where our method outperforms all baselines in different observation percentages. Figure 2 (a) illustrates the benefit of the CSSC over cubic spline interpolation and ODE-RNN. Cubic spline interpolation cannot interpolate the curve when observations are sparse, without learning from the dataset. ODE-RNN performs poorly when a jump occurs at the observation. However, CSSC can help eliminate such jump and guarantee smoothness at the observation time, thus yielding good interpolation. In Figure 2 (b), we visualized the ODE-RNN output $o$ and compensation $c$, and the CSSC output $\hat{o}$ in detail to demonstrate how the the bad $o$ becomes a good $\hat{o}$ by adding $c$. Finally, Fig. 2 (c) demonstrates the first dimension of hidden states before and after smoothing, where the smoothed hidden state leads to a smoothed output.

Table 1: Interpolation MSE on toy, MuJoCo, Moving MNIST test sets with different percentages of observations.

| Observation | Toy | | | MuJoCo | | | Moving MNist | |
|---|---|---|---|---|---|---|---|---|
| | 10% | 30% | 50% | 10% | 30% | 50% | 20% | 30% |
| Cubic Spline | 0.801249 | 0.003142 | 0.000428 | 0.016417 | 0.000813 | 0.000125 | 0.072738 | 0.252647 |
| RNN-$\Delta t$ | 0.449091 | 0.245546 | 0.102043 | 0.028457 | 0.019682 | 0.008975 | 0.040392 | 0.037924 |
| GRU-D | 0.473954 | 0.247522 | 0.121482 | 0.056064 | 0.018285 | 0.008968 | 0.037503 | 0.035681 |
| Latent-ODE | **0.013768** | 0.002282 | 0.002031 | 0.010246 | 0.009601 | 0.009032 | 0.035167 | 0.031287 |
| GRU-ODE-Bayes | 0.117258 | 0.010716 | 0.000924 | 0.028457 | 0.006782 | 0.002352 | 0.034093 | 0.030975 |
| ODE-RNN | 0.025336 | **0.001429** | 0.000441 | 0.011321 | 0.001572 | 0.000388 | 0.022141 | 0.016998 |
| hidden CSSC | 0.027073 | 0.001503 | 0.000244 | **0.004554** | **0.000378** | **0.000106** | **0.019475** | **0.015662** |
| CSSC | **0.024656** | **0.000457** | **0.000128** | **0.006097** | **0.000375** | **0.000087** | - | - |

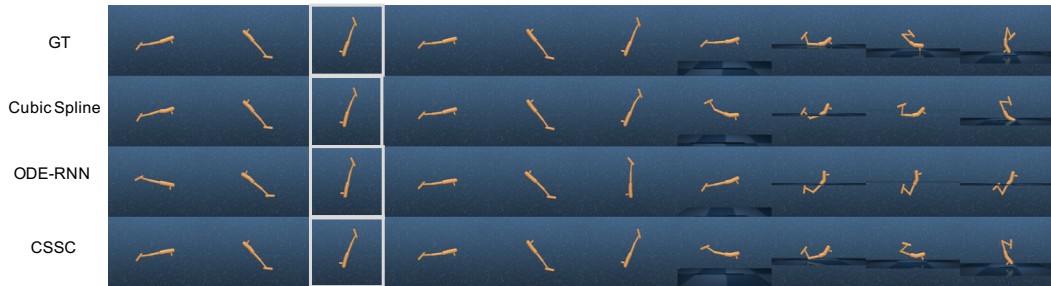

Figure 3: The visual result for MuJoCo. We visualize part of the interpreted trajectory with 5 frames interval. 10% frames are observed out of 100 frames. The observation is highlighted with white box.

### 4.3 MuJoCo Physics Simulation

We test our proposed method with the "hopper" model provided by DeepMind Control Suite (Tassa et al., 2018) based on MuJoCo physics engine. To increase the trajectory's complexity, the hopper is thrown up, then rotates and freely falls to the ground (Fig. 3). We will interpolate the 7-dimensional state that describes the position of the hopper. Both our hidden CSSC and CSSC achieve improved performance from ODE-RNN, especially when observations become sparser, as Tab. 1 indicates. The visual result is shown in Fig. 3, where CSSC resemble the GT trajectory most.

### 4.4 Moving MNIST

In addition to low-dimensional data, we further evaluate our method on high-dimensional image interpolation. Moving MNIST consists of 20-frame video sequences where two handwritten digits are drawn from MNIST and move with arbitrary velocity and direction within the 64×64 patches, with potential overlapping and bounce at the boundaries. As a matter of expediency, we use a subset of 10k videos and resize the frames into 32×32. 4 (20%) and 6 (30%) frames out of 20 are randomly observed, including the starting and ending frames. We encode the image with 2 ResBlock (He et al., 2016) into 32-d hidden vector and decode it to pixel space with a stack of transpose convolution layers. Since the pixels are only continuous at the semantic level, only hidden CSSC is evaluated with comparison methods. As shown in Tab. 1, the hidden CSSC can further improve ODE-RNN's result, and spline interpolation behaves the worse since it can only interpolate at the pixel space, which is discontinuous through time. The visual result (Fig. 4) shows that the performance gain comes from the smoother movement and the clearer overlapping.

### 4.5 Ablation Study

**The effect of end-to-end training**. Apart from our standard end-to-end training of CSSC, two alternative training strategies are pre-hoc and post-hoc CSSN. Pre-hoc CSSC is to train a standard CSSC but only use the ODE-RNN part when inference. On the contrary, post-hoc CSSC is to train an ODE-RNN without CSSC, but apply CSSC upon the output of ODE-RNN when inference. The

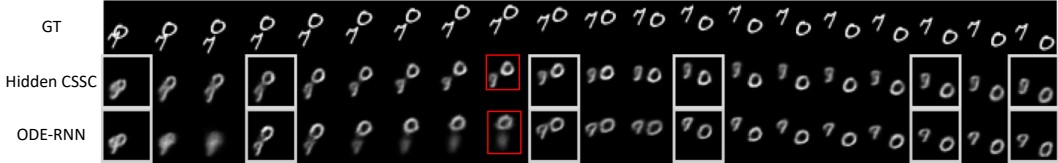

Figure 4: The visual result for Moving MNIST. The observation is indicated in white box. The comparison of discontinuity is highlighted in red box.

Table 2: The MSE for on MuJoCo test set for the study of different training strategies.

|  | 10% | 30% | 50% |
|---|---|---|---|
| ODE-RNN | 0.011321 | 0.001572 | 0.000388 |
| pre-hoc CSSC | 0.053217 | 0.006131 | 0.002062 |
| post-hoc CSSC | 0.013574 | 0.000514 | 0.000110 |
| CSSC | **0.006097** | **0.000375** | **0.000087** |

comparison of pre-hoc, post-hoc, and standard CSSC is presented in Tab. 2, where standard CSSC behaves the best. The pre-hoc CSSC is the worst because training with CSSC can tolerate the error of the ODE-RNN; thus, inference without CSSC exposes the error of ODE-RNN, and it even performs worse than standard ODE-RNN. The post-hoc CSSC can increase the performance of ODE-RNN with simple post-processing when inference. However, such performance gain is not guaranteed when the observation is sparse. For example, the post-hoc CSSC even decreases the performance of ODE-RNN in 10% observation setting. Standard CSSC has higher performance than ODE-RNN in all our experiments, indicating the importance of end-to-end training.

**The effect of $\alpha$.** We study the effect of $\alpha$ ranging from 0 to 10000 given different percentages of observation on MuJoCo dataset. The performance of CSSC is quite robust to the choice of $\alpha$, shown in Tab. 4 (in Appendix), especially the MSE only fluctuated from 0.000375 to 0.000463 as $\alpha$ ranges from 1 to 10000 in 30% observation setting. Interestingly, Fig. 5 (in Appendix) visually compares the interpolation for $o$, $c$, and $\hat{o}$ under variant $\alpha$ and indicates higher $\alpha$ contributes to lower $c$, thus $o$ is more dominant of smoothed output $\hat{o}$. On the other hand, the smaller $\alpha$ will make $o$ less correlated with the ground truth, but the CSSC $c$ can always make $\hat{o}$ a well-interpolated curve.

## 5 DISCUSSION

**Limitations**. While the CSSC model interpolates the trajectory that can strictly cross the observation points, such interpolation is not suitable for noisy data whose observations are inaccurate. Moreover, interpolation error bound (Eq. (15)) requires the underlying data is fourth-order continuous, which indicates that CSSC is not suitable to interpolate sharply changed data, e.g., step signals.

**Future work**. The CSSC can not only be applied to ODE-RNN but can smooth any piecewise continuous function. Applying CSSC to other more general models is a desirable future work. Also, while interpolation for noisy data is beyond this paper's scope, but hidden CSSC shows the potential to tolerate data noise by capturing the data continuity at the semantic space rather than the observation space, which can be a future direction.

## 6 CONCLUSION

We introduce the CSSC that can address the discontinuity issue for ODE-RNN. We have derived the analytical solution for the CSSC and even proved its error bound for the interpolation task, which is hard to obtain in pure neural network models. The CSSC combines the modeling ability of deep neural networks (ODE-RNN) and the smoothness advantage of cubic spline interpolation. Our experiments have shown the benefit of such a combination. The hidden CSSC extends the smoothness from output space to the hidden semantic space, enabling a more general format of continuous signals.

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

## A  PROOF OF INTERPOLATION

Substitute $\hat{o}$ with Eq. (5) we will have

$$c_k(t_k^+) + o(t_k^+) = x_k \tag{18}$$

$$c_k(t_{k+1}^-) + o(t_{k+1}^-) = x_{k+1} \tag{19}$$

$$\dot{c}_{k-1}(t_k^-) + \dot{o}(t_k^-) = \dot{c}_k(t_k^+) + \dot{o}(t_k^+) \tag{20}$$

$$\ddot{c}_{k-1}(t_k^-) + \ddot{o}(t_k^-) = \ddot{c}_k(t_k^+) + \ddot{o}(t_k^+) \tag{21}$$

$$\ddot{c}_0(t_0^+) + \ddot{o}(t_0^+) = 0 \tag{22}$$

$$\ddot{c}_{n-1}(t_n^-) + \ddot{o}(t_n^-) = 0 \tag{23}$$

And we let $\ddot{c}_k(t_k^+) = M_k, k = 0, 1, ..., n-1$, and $\ddot{c}_{n-1}(t_n^-) = M_n$. We define the first order and second order jump difference of ODE-RNN as

$$\dot{r}_k = \dot{o}(t_k^+) - \dot{o}(t_k^-); \tag{24}$$

$$\ddot{r}_k = \ddot{o}(t_k^+) - \ddot{o}(t_k^-). \tag{25}$$

With Eq. (21), we have

$$\hat{M}_{k+1} = \ddot{c}_k(t_{k+1}^-) = M_{k+1} + \ddot{r}(t_{k+1}). \tag{26}$$

Using constraint Eq. (18)(19), we denote

$$c_k(t_k) = x_k - o(t_k^+) = \epsilon_k^+ \tag{27}$$

$$c_k(t_{k+1}) = x_{k+1} - o(t_{t+1}^-) = \epsilon_{k+1}^-. \tag{28}$$

Also denote the step size $\tau_k = t_{k+1} - t_k$. Then applying constraint Eq. (18)(19)(21) we have the piece cubic function expressed as

$$c_k(t) = \frac{\hat{M}_{k+1} - M_k}{6\tau_k}(t-t_k)^3 + \frac{M_k}{2}(t-t_k)^2 + \left(\frac{\epsilon_{k+1}^- - \epsilon_k^+}{\tau_k} - \frac{\tau_k(\hat{M}_{k+1} + 2M_k)}{6}\right)(t-t_k) + \epsilon_k^+. \tag{29}$$

Next, we try to solve all $M_k$. We firstly express $\dot{c}_{k-1}(t_k^-)$ and $\dot{c}_k(t_k^+)$ as

$$\dot{c}_{k-1}(t_k^-) = \frac{\hat{M}_k - M_{k-1}}{2}\tau k - 1 + M_{k-1}\tau_{k-1} + \frac{\epsilon_k^- - \epsilon_{k-1}^+}{\tau_{k-1}} - \frac{\hat{M}_k + 2M_{k-1}}{6}\tau_{k-1}, \tag{30}$$

$$\dot{c}_k(t_k^+) = \frac{\epsilon_{k+1}^- - \epsilon_k^+}{\tau_k} - \frac{\hat{M}_{k+1} + 2M_k}{6}\tau_k \tag{31}$$

Applying Eq. (20) we have

$$2M_k + \frac{\tau_{k-1}}{\tau_{k-1} + \tau_k} M_{k-1} + \frac{\tau_k}{\tau_{k-1} + \tau_k} M_{k+1} = 6 \frac{\epsilon[t_k^+, t_{k+1}^-] - \epsilon[t_{k-1}^+, t_k^-]}{\tau_{k-1} + \tau_k} + \frac{6\dot{r}_k - 2\ddot{r}_k \tau_{k-1} - \ddot{r}_{k+1} \tau_k}{\tau_{k-1} + \tau_k}$$
(32)

where $\epsilon[t_k^+, t_{k+1}^-] = \frac{\epsilon_{k+1}^- - \epsilon_k^+}{\tau_k}$.

$$\mu_k = \frac{\tau_{k-1}}{\tau_{k-1} + \tau_k}$$
(33)

$$\lambda_k = \frac{\tau_k}{\tau_{k-1} + \tau_k}$$
(34)

$$d_k = 6 \frac{\epsilon[t_k^+, t_{k+1}^-] - \epsilon[t_{k-1}^+, t_k^-]}{\tau_{k-1} + \tau_k} + \frac{6\dot{r}_k - 2\ddot{r}_k \tau_{k-1} - \ddot{r}_{k+1} \tau_k}{\tau_{k-1} + \tau_k}.$$
(35)

Then $M_k$ can be obtained by solving by system of linear equations:

$$A\mathbf{M} = \mathbf{d},$$
(36)

where

$$A = \begin{pmatrix} 2 & \lambda_1 & & & \\ \mu_2 & 2 & \lambda_2 & & \\ & \ddots & \ddots & \ddots & \\ & & \mu_{n-2} & 2 & \lambda_{n-2} \\ & & & \mu_{n-1} & 2 \end{pmatrix}, \mathbf{M} = \begin{pmatrix} M_1 \\ M_2 \\ \vdots \\ M_{n-2} \\ M_{n-1} \end{pmatrix}, \mathbf{d} = \begin{pmatrix} d_1 \\ d_2 \\ \vdots \\ d_{n-2} \\ d_{n-1} \end{pmatrix}$$
(37)

And $A$ is non-singular since it is strict diagonally dominant matrix, hence guarantee single solution for $M_k$. Hence

$$\mathbf{M} = A^{-1}\mathbf{d}.$$
(38)

Now we still need to calculate $\dot{o}(t)$ and $\ddot{o}(t)$.

$$\dot{o}(t) = \frac{\partial g}{\partial \mathbf{h}}^\mathsf{T} \frac{\partial \mathbf{h}}{\partial t} = \frac{\partial g}{\partial \mathbf{h}}^\mathsf{T} f$$
(39)

$$\dot{\mathbf{h}}(t) = \frac{df(\mathbf{h}(t))}{dt} = \frac{\partial f}{\partial \mathbf{h}}^\mathsf{T} \frac{d\mathbf{h}}{dt} = \frac{\partial f}{\partial \mathbf{h}}^\mathsf{T} f,$$
(40)

$$\ddot{o}(t) = \frac{d\frac{\partial g}{\partial \mathbf{h}}^\mathsf{T}}{dt} \dot{\mathbf{h}} + \frac{\partial g}{\partial h}^\mathsf{T} \ddot{\mathbf{h}} = (\frac{\partial^2 g}{\partial \mathbf{h}^2} f)^\mathsf{T} f + \frac{\partial g}{\partial \mathbf{h}}^\mathsf{T} \frac{\partial f}{\partial \mathbf{h}} f$$
(41)

## B  PROOF FOR ERROR BOUND

The following proof are based on the default notation and setting for scalar time series interpolation: given a set of $n+1$ points $\{x(t_i)\}_{i=0}^n$ irregularly spaced at time points $\Pi : a = t_0 < t_1 < ... < t_n = b$, the goal is to approximate the underlying ground truth function $x(t)$, $t \in \Omega = [a, b]$. Let $o(t)$ as the ODE-RNN prediction, and $\hat{o}(t) = c(t) + o(t)$ as our smoothed output where $c(t)$ is the compensation defined by Eq. (11). To simplify the notation, we drop the argument $t$ for a function, e.g. $x(t) \to x$. We firstly introduce several lemmas, then come the the main proof for the interpolation error bound.

**Lemma 4.** *(Hall & Meyer, 1976) Let $e$ be any function in $C^2(\Pi) \bigcap_{k=1}^n C^4(t_{k-1}, t_k)$ with constraints $x(t_k) = 0$, $k = 0, ..., n$, and natural boundary $e''(a) = e''(b) = 0$, then*

$$|e^{(r)}(t_k)| \leq \rho_r ||e^{(4)}||_\infty \tau^{4-r} \qquad (k = 0, ..., n; \ r = 1, 2),$$
(42)

*where $\rho_1 = \frac{1}{24}$, $\rho_2 = \frac{1}{4}$.*

**Lemma 5.** *Let $e = x - \hat{o}$, if $x \in C^4(\Omega)$, $f \in C^3$, $g \in C^4$, then $e \in C^2(\Pi) \bigcap_{k=1}^n C^4(t_{k-1}, t_k)$*

*Proof.* Since $\hat{o} \in C^2(\Pi)$, and $c \in \bigcap_{k=1}^n C^4(t_{k-1}, t_k)$, hence we only have to prove that $o \in \bigcap_{k=1}^n C^4(t_{k-1}, t_k)$. Since $o(t)$ in each time period $(t_{k-1}, t_k)$ is calculated by ODE (Eq. (1), and (2)), we can express different orders of the derivative of $o$ as:

$$\dot{o} = \frac{\partial g}{\partial \mathbf{h}}^\mathsf{T} f,$$
(43)

$$\ddot{o} = (\frac{\partial^2 g}{\partial \mathbf{h}^2} f)^\mathsf{T} f + \frac{\partial g}{\partial \mathbf{h}}^\mathsf{T} \frac{\partial f}{\partial \mathbf{h}} f,$$
(44)

$$o^{(3)} = f^{\mathsf{T}} \left( \frac{\partial f}{\partial \mathbf{h}} \frac{\partial^2 g}{\partial \mathbf{h}^2} + \frac{\partial^3 g}{\partial \mathbf{h}^3} \circ f + 2 \frac{\partial^2 g}{\partial \mathbf{h}^2} \frac{\partial f}{\partial \mathbf{h}} \right) f + \frac{\partial g}{\partial \mathbf{h}}^{\mathsf{T}} \left( \frac{\partial^2 f}{\partial \mathbf{h}^2} \circ f + \frac{\partial f}{\partial \mathbf{h}} \frac{\partial f}{\partial \mathbf{h}} \right) f, \tag{45}$$

$$\begin{aligned} o^{(4)} = &f^{\mathsf{T}} \Big( \frac{\partial^4 g}{\partial \mathbf{h}^4} \circ f \circ f + \frac{\partial^3 g}{\partial \mathbf{h}^3} \circ \left( \frac{\partial f}{\partial \mathbf{h}} \frac{\partial f}{\partial \mathbf{h}} \right) + 3 \frac{\partial^3 g}{\partial \mathbf{h}^3} \circ f \frac{\partial f}{\partial \mathbf{h}} + 2 \frac{\partial f}{\partial \mathbf{h}} \frac{\partial^3 g}{\partial \mathbf{h}^3} \circ f + 3 \frac{\partial^2 g}{\partial \mathbf{h}^2} \frac{\partial^2 f}{\partial \mathbf{h}^2} \circ f \\ &+ 3 \frac{\partial^2 g}{\partial \mathbf{h}^2} \frac{\partial f}{\partial \mathbf{h}} \frac{\partial f}{\partial \mathbf{h}} + \frac{\partial^2 f}{\partial \mathbf{h}^2} \circ f \frac{\partial^2 g}{\partial \mathbf{h}^2} + \frac{\partial f}{\partial \mathbf{h}} \frac{\partial f}{\partial \mathbf{h}} \frac{\partial^2 g}{\partial \mathbf{h}^2} + 3 \frac{\partial f}{\partial \mathbf{h}} \frac{\partial^2 g}{\partial \mathbf{h}^2} \frac{\partial f}{\partial \mathbf{h}} \Big) f \\ &+ \frac{\partial g}{\partial \mathbf{h}}^{\mathsf{T}} \Big( \frac{\partial^3 f}{\partial \mathbf{h}^3} \circ f \circ f + \frac{\partial^2 f}{\partial \mathbf{h}^2} \circ \left( \frac{\partial f}{\partial \mathbf{h}} f \right) + 2 \frac{\partial^2 f}{\partial \mathbf{h}^2} \circ f \frac{\partial f}{\partial \mathbf{h}} + \frac{\partial f}{\partial \mathbf{h}} \frac{\partial^2 f}{\partial \mathbf{h}^2} \circ f + \frac{\partial f}{\partial \mathbf{h}} \frac{\partial f}{\partial \mathbf{h}} \frac{\partial f}{\partial \mathbf{h}} \Big) f, \end{aligned} \tag{46}$$

where $\frac{\partial f}{\partial \mathbf{h}}$ is the Jacobian matrix since $f$ is a multi-valued function, $\frac{\partial^2 g}{\partial \mathbf{h}^2}$ is the Hessian matrix since $o$ is a scalar and $g$ is single-valued function. The higher order derivative than second order (Hessian) becomes multi-dimensional matrix which can not be mathmatically expressed for standard matrix production, so we indicate the product of a multi-dimensional matrix and a vector as $\circ$, which has higher computing priority then normal matrix production. From Eq. (46), $o^{(4)}$ depends on $\frac{\partial^4 g}{\partial \mathbf{h}^4}$ and $\frac{\partial^3 f}{\partial \mathbf{h}^3}$. Hence given $g \in C^4$ and $f \in C^3$, we obtain $o \in \bigcap_{k=1}^{n} C^4(t_{k-1}, t_k)$.

$\square$

**Lemma 6.** *(Birkhoff & Priver, 1967) Given any function $v \in C^4(t_k, t_k + 1)$, let $u$ be the cubic Hermite interpolation matching $v$, we have*

$$|(v(t) - u(t))^{(r)}| \leq A_r(t) ||v^{(4)}||_\infty \tau^{4-r} \qquad (r = 0, 1), \tag{47}$$

*with*

$$A_r(t) = \frac{\tau_i^r [(t - t_i)(t_{i+1} - t)]^{2-r}}{r!(4 - 2r)!\tau^{4-r}} \qquad \text{if } t \in [t_i, t_j). \tag{48}$$

*This is the error bound for cubic Hermite interpolation.*

Given the above lemmas, we are ready to the formal proof for Theorem 2.
**Proof of Theorem 2:**

*Proof.* According to Lemma 6, we let $v = x - o$ (since $x - o \in C^4(t_k, t_{k+1})$ according to Lemma 5), let $u$ be the cubic Hermite interpolation matching $v$, then we have

$$|(x(t) - o(t) - u(t))^{(r)}| \leq A_r(t) ||(x - o)^{(4)}||_\infty \tau^{4-r} \qquad (r = 0, 1), \tag{49}$$

with

$$A_r(t) = \frac{\tau_i^r [(t - t_i)(t_{i+1} - t)]^{2-r}}{r!(4 - 2r)!\tau^{4-r}} \qquad \text{if } t \in [t_i, t_j). \tag{50}$$

Let $e = x - \hat{o}$, $\varepsilon = u - c$, then $\varepsilon$ is cubic Hermite interpolation implying

$$\varepsilon(t_k) = x(t_k) - o(t_k) - c(t_k) = x(t_k) - \hat{o}(t_k) = 0, \tag{51}$$

and

$$\dot{\varepsilon}(t_k) = \dot{x}(t_k) - \dot{o}(t_k) - \dot{c}(t_k) = \dot{x}(t_k) - \dot{\hat{o}}(t_k) = \dot{e}(t_k). \tag{52}$$

Therefore, $\varepsilon(t)$ in $[t_k, t_{k+1})$ is a cubic function with endpoints satisfying can be constructed for each interval $[t_k, t_{k+1})$ as Eq. (51) and (52), which can be reconstructed as

$$\varepsilon(t) = \dot{e}(t_k) H_1(t) + \dot{e}(t_{k+1}) H_2(t), \tag{53}$$

where

$$H_1(t) = (t - t_k)(t - t_{k+1})^2 / \tau_k^2, \tag{54}$$

$$H_2(t) = (t - t_{k+1})(t - t_k)^2 / \tau_k^2. \tag{55}$$

From Lemma 4 and Lemma 5, $e$ is bounded as Eq. (42). Combining Eq. (42) and (53) yields:

$$|\varepsilon^{(r)}(t)| \leq \rho_1 ||e^{(4)}||_\infty \left( |H_1^{(r)}(t)| + |H_2^{(r)}(t)| \right) \tau^3 \qquad (r = 0, 1), \tag{56}$$

Table 3: The interpolation MSE for on toy sinuous wave test set at different observation ratio. We compare the analytical and numerical differentiation of $\dot{o}$ and $\ddot{o}$ for CSSC under different settings, where *block* means block gradient, *drop* indicates set as zero, and *CSSC* is the standard implementation.

| | Analytical | | | Numerical | | |
|---|---|---|---|---|---|---|
| Observation | 10% | 30% | 50% | 10% | 30% | 50% |
| Block $\dot{o}$, $\ddot{o}$ | 10.601951 | 0.001408 | 0.000150 | - | - | - |
| Block $\dot{o}$ | 0.811334 | 0.000721 | 0.000142 | - | - | - |
| Block $\ddot{o}$ | 0.406867 | 0.000348 | 0.000121 | - | - | - |
| Drop $\ddot{o}$ | 0.072519 | 0.000356 | 0.000123 | - | - | - |
| CSSC | **0.024656** | **0.000457** | **0.000128** | **0.067881** | **0.000397** | **0.000126** |

which is rewritten as

$$|(x(t) - \hat{o}(t))^{(r)}| \leq B_r(x)\|e^{(4)}\|_\infty \tau^{4-r} = B_r(x)\|(x-o)^{(4)}\|_\infty \tau^{4-r}, \tag{57}$$

with $B_r(t) = \rho_1\big(|H_1^{(r)}(t)| + |H_2^{(r)}(t)|\big)\tau^{r-1}$.

Using Triangle inequality, Lemma 6, and Eq. 57, it issues

$$
\begin{aligned}
|x^{(r)}(t) - \hat{o}^{(r)}(t)| &= |x^{(r)}(t) - o^{(r)}(t) - u^{(r)}(t) + u^{(r)}(t) - c^{(r)}(t)| \\
&\leq |x^{(r)}(t) - o^{(r)}(t) - u^{(r)}(t)| + |u^{(r)}(t) - c^{(r)}(t)| \\
&\leq (A_r(x) + B_r(x))\|(x-o)^{(4)}\|_\infty \tau^{4-r}. \tag{58}
\end{aligned}
$$

Let $C_r(x) = A_r(x) + B_r(x)$, and an analysis of the optimality (Hall & Meyer, 1976) yields

$$C_0(x) \leq \frac{5}{384}; \qquad C_1(x) \leq \frac{1}{12}. \tag{59}$$

$\square$

## C   COMPUTATIONAL COMPLEXITY

### C.1   IMPLEMENTATION OF THE INVERSE OF MATRIX

For sake of simplicity, our Pytorch implementation adopts `torch.inverse` to compute $A^{-1}$ in Eq. (12), which is actually the implementation of the LU composition using partial pivoting with best complexity $O(n^2)$. Its complexity is higher than the complexity of tridiagonal matrix algorithm $O(n)$, whose implementation will be left for future work.

### C.2   COMPUTATION OF $\dot{o}$ AND $\ddot{o}$

As Eq. 8 indicates, computing $\dot{o}$ and $\ddot{o}$ requires the Hessian and Jacobian of $g$ and the Jacobian of $f$. These Jacobians and Hessians will first participate in the inference stage and then are involved in the gradient backpropagation. However, the latest Pytorch[1] does not support the computing of Jacobian and Hessian in batch, so the training process will be very slow in practice, rendering the computing cost prohibitively high. Therefore, we propose two ways to circumvent such issues from both numerical and analytical views.

---

[1]Pytorch Version 1.6.0, updated in July 2020.

**Numerical Differentiation**. The first solution is to use a numerical derivative. We approximate the left limitation and right limitation of $\dot{o}(t)$ and $\ddot{o}(t)$ as

$$\dot{o}(t^-) = \frac{o(t^-) - o(t - \Delta t)}{\Delta} \tag{60}$$

$$\ddot{o}(t^-) = \frac{o(t^-) - 2o(t - \Delta) + o(t - 2\Delta)}{\Delta^2} \tag{61}$$

$$\dot{o}(t^+) = \frac{o(t + \Delta t) - o(t^+)}{\Delta} \tag{62}$$

$$\ddot{o}(t^+) = \frac{o(t + 2\Delta t) - 2o(t + \Delta t) + o(t^+)}{\Delta^2} \tag{63}$$

where $\Delta = 0.001$. In this way, we can avoid computing Jacobian or Hessian, and the computational complexity almost remains the same as ODE-RNN. The last row of Table. 3 shows that the numerical differentiation can maintain the same performance with the analytical solution with 30% and 50% observation. However, when observations become sparser, e.g., 10%, the analytical differentiation will gain a better performance.

**Analytical Approximation**. For analytical solution, the major computing burden is the Hessian matrix; thus, we approximate $\ddot{o}$ with Eq. (65) by simply dropping the Hessian term. The motivation and rationality is detailed in Appx. C.3.

### C.3 COMPUTATION REDUCTION FOR ANALYTICAL DERIVATIVE

To further reduce the computation for the analytical derivative of $\dot{o}$ and $\ddot{o}$, we investigate whether blocking the gradient of $\dot{o}$ or $\ddot{o}$ will affect the interpolation performance. From first three rows in Table. 3, we can see that performance of blocking the gradient of $\dot{o}$ is worse than that of blocking $\ddot{o}$, indicating $\dot{o}$ is more important than $\ddot{o}$ in terms of computing the compensation $\mathbf{c}$. In light of this, we further drop $\ddot{o}$ in the Eq. (7), meaning the second order jump difference $\ddot{r}_k$ is zero. According to the performance shown in the fourth row of Table. 3, $\ddot{o}$ has a minor impact on the compensation $c$ when the observation is dense. We investigate the reason by check how $\ddot{o}$ impact the computation of $c$ and find that they are correlated by $d_k$ in Eq. (12). We write $d_k$ again here for better clarification:

$$d_k = 6\frac{\epsilon[t_k^+, t_{k+1}^-] - \epsilon[t_{k-1}^+, t_k^-]}{\tau_{k-1} + \tau_k} + \frac{6\dot{r}_k - 2\ddot{r}_k\tau_{k-1} - \ddot{r}_{k+1}\tau_k}{\tau_{k-1} + \tau_k}. \tag{64}$$

From (Eq. (6, 7)) and second term of above equation (Eq. (64)), we noticed that $\dot{r}_k, \ddot{r}_k$ serve as the bridge between $\dot{o}$ and $\ddot{o}$ and $d_k$, indicating that the importance of $\dot{o}$ and $\ddot{o}$ in generating compensation $c$ can be examined by estimating the relative significance of $\dot{r}_k, \ddot{r}_k$ appear in $d_k$. We can denote the relative significance as fraction of terms included $\ddot{r}_k$ and $\dot{r}_k$ as $s = \frac{|2\ddot{r}_k\tau_{k-1} + \ddot{r}_{k+1}\tau_k|}{|6\dot{r}_k|} \leq \frac{3\max\{|\ddot{r}_k\tau_{k-1}|, |\ddot{r}_{k+1}\tau_k|\}}{6|\dot{r}_k|}$. Without loss of generality, we let $\max\{|\ddot{r}_k\tau_{k-1}|, |\ddot{r}_{k+1}\tau_k|\} = |\ddot{r}_k\tau_{k-1}|$, then we have $s \leq \frac{3\max\{|\ddot{r}_k\tau_{k-1}|, |\ddot{r}_{k+1}\tau_k|\}}{6|\dot{r}_k|} = \frac{|\ddot{r}_k\tau_{k-1}|}{2|\dot{r}_k|}$. It shows that $\ddot{r}_k$ has a coefficient of $\tau_{k-1}$ on the numerator, which is the time interval between two adjacent observations. In our experiment, 100 samples is uniformly sampled in 5s, and a certain percent of the samples will be selected as observations. In this setting, if we have 50% samples as observations, the average $\tau_k = 5/50 = 0.1$; thus $s$ can be informally estimated as $s \leq \frac{|\ddot{r}_k|}{20|\dot{r}_k|}$, indicating $\dot{o}$ is more important than $\ddot{o}$ in our experiments. As the observation ratio is higher, the $\tau_k$ becomes smaller, hence the relative significance of $\ddot{r}_k$ and $\dot{r}_k$ will become even larger. Armed with the above intuition that $\ddot{o}$ is less important and the fact that he Hessian contribute the major complexity $O(W^2)$, we drop the term with Hessian and find a better approximation of $\ddot{o}$, leading the final approximation of $\ddot{o}$ as:

$$\ddot{o} = \frac{\partial g}{\partial \mathbf{h}}^{\mathsf{T}} \frac{\partial f}{\partial \mathbf{h}} f. \tag{65}$$

Such approximation yields descent performance in practice. In addition, because $\mathbf{o}$ is multi-variable and $g$ is a multi-valued function in practice, and such Jacobian needs to run in batch. We implement our Jacobian operation to tackle these difficulties and can run fast.

Table 4: The MSE of CSSC for different $\alpha$ on MuJoCo test set. It compares the different data samples.

| $\alpha$ | 0 | 1 | 10 | 100 | 1000 | 10000 |
|---|---|---|---|---|---|---|
| 10% | 0.006551 | 0.011915 | 0.009691 | **0.005421** | 0.006097 | 0.005886 |
| 30% | 0.000745 | 0.000463 | 0.000426 | 0.000400 | **0.000375** | 0.000422 |
| 50% | 0.000126 | 0.000102 | 0.000074 | 0.000072 | 0.000087 | **0.000057** |

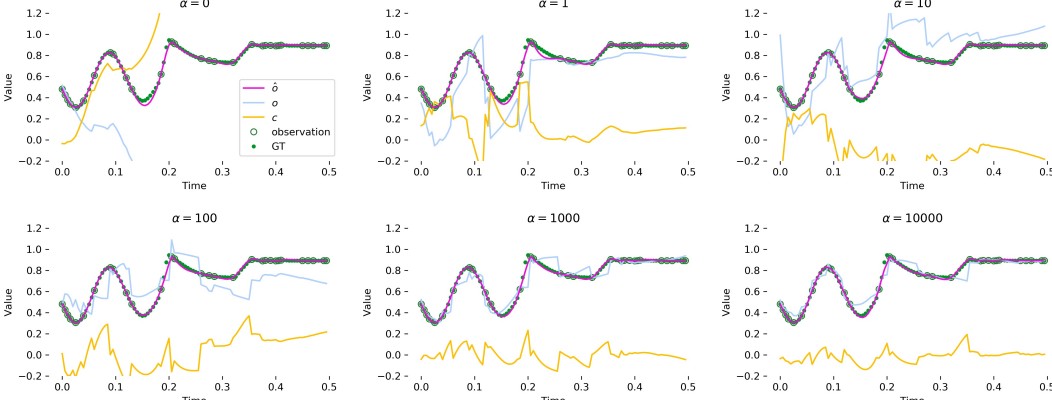

Figure 5: The interpolation MSE of CSSC with different $\alpha$ of MuJoCo dataset. The curve is the 4-th dimension of the hopper's state.

## D  IMPLEMENTATION DETAILS

The neural ODE state $\mathbf{h}$ has size 15. $f$ is a 5-layer MLP with hidden state size 300. $g$ is a 2-layer MLP with hidden state size 300. RNNCell has hidden state size 100. To obey the condition $f \in C^3$ and $g \in C^4$ required by the error bound in Theorem 2, we select all the nonlinear function as tanh function. The $\alpha$ is selected as 1000 based on the ablation study of $\alpha$ in Sec. 4.5. The network is optimized by AdaMax (Kingma & Ba, 2014) with $\beta_1 = 0.9$, $\beta_2 = 0.999$, and learning rate 0.02.

