# OpenReview forum: "Cubic Spline Smoothing Compensation for Irregularly Sampled Sequences"
_ICLR.cc/2021/Conference — Reject_

### Official Review · AnonReviewer1 · 2020-10-25
**A nice solution to insufficiency in neural differential equation by introducing the cubic spline smoothing compensation**

**Rating:** 7
**Confidence:** 3

**Review:**

This paper presented how cubic spline smoothing function was used to compensate the insufficiency in ODE-RNN (ordinary differential equation recurrent neural network) for irregularly sampled sequences.

Pros:
This paper presented a nice solution to enhance the performance of ODE-RNN with mathematical evidence. Theoretical justification was provided. The solution was evaluated in different tasks.

Cons:
This solution required the calculation of inverse matrix in a compensation term which would cause a significant increase in training time when the dimension is enlarged.

---

> ### Author Response · Authors · 2020-11-25
> **Response to R1**
>
> Thanks for acknowledging our efforts in this paper.
>
> **Calculation of the inverse matrix**: as it is a tridiagonal matrix, it can be computed in $O(n)$, which is already much faster than standard matrix inverse $O(n^2)$. Therefore, there is no significant increase in computation as the dimension $n$ increases. Moreover, if the data dimension $n$ is very high, hidden CSSC is a good choice where only the fixed hidden state dimension needs computation of the matrix inverse。

---

### Official Review · AnonReviewer4 · 2020-10-26

**Rating:** 5
**Confidence:** 5

**Review:**

Summary: This work addresses the discontinuity issues caused by jumps in hidden state/output at the arrival of new observations in a ODE-RNN. This problem is tackled by adding a cubic spline smoothing component on top of ODE-RNN to produce smooth and continuous hidden state/outputs. They derive a closed form solution for the cubic spline component based on the output of ODE-RNN and obtain an error bound for 4th order derivable inputs. Although the cubic spline component has no trainable component, this work shows that the gradient can flow through it to ODE-RNN and perform end-to-end training.

Positives:
1. The paper focus on the task of learning from irregularly sampled data which is important in many domains.
2. The paper is well written and easy to follow.
3. Experiments show that the model indeed learns to perform continuous interpolation.

Concerns:
1. The key concern about the paper is the lack of rigorous experimentation to study the usefulness of the proposed method. The authors mention that this approach would be most useful in continuous time series interpolation for irregularly sampled time series , but there have been no experiments on real world irregularly sampled time series (e.g. PhysioNet).
2. Another concern I have is with the performance of Latent ODE baseline. The results in Rubanova et al. show that Latent-ODE model almost always outperforms ODE-RNN but that is not the conclusion I get from the results in this paper. This makes me question the validity of implementation/experiments.
3. ODE-RNN can also be used for extrapolation in addition to interpolation. Is it possible to perform extrapolation using the proposed model?

Additional Comments:
1. Could the authors comment on the performance of a Latent ODE model with CSSC as encoder and ODE as decoder?
2. It would be interesting to see if the improved interpolation leads to improved performance in downstream tasks such as sequence classification.
2. Missing comparison with kernel based interpolation methods e.g. Shukla and Marlin (2019).

---

> ### Author Response · Authors · 2020-11-25
> **Response to R4**
>
> Thanks for R4's valuable opinions
>
> **Real Dataset**: As our major contribution is focused on the construction, computation, and error analysis of CSSC on unnoisy time series, we have validated the effect of our method on three unnoisy datasets. Some other datasets e.g. PhysioNet that contain noisy data are beyond the scope of this paper.
> A thorough investigation for noisy data could be our future step as our hidden CSSC has the potential to cope with noisy time series.
>
> **Latent ODE is weaker than ODE-RNN**. We suppose that in the long time series the latent ODE may not be strong enough to remember all the observations by just a latent state, but ODE-RNN does not have this issue.
>
> **Is it able to do extrapolation?** Our proposed method addresses the limitation of the ODE-RNN model that the trajectory is continuous everywhere except the observation points. Since extrapolation will not meet new observation, the extrapolated trajectory is always continuous; therefore we suspect that directly using our method may not bring a large gain over the latent ODE at the extrapolation task. However, our model is still important to inspire how the latent state of the whole sequence is encoded given the hidden state smoothed at every point.
>
> **A Latent ODE model with CSSC as encoder and ODE as decoder**. Thanks for this suggestion where the combination of ODE-RNN+CSSC encoder and ODE decoder can even support the extrapolation task. However, if we use the hidden CSSC to smooth the hidden state of ODE-RNN, the final step state will not be changed, and only the hidden states between the observation time gap are changed. Therefore, since the encoder only outputs the final hidden state, the hidden CSSC does not differ from ODE-RNN as an encoder. And another way is to use CSSC for data imputing and then use whatever encoder -- even an RNN -- to encode the sequence into a hidden state. We think the second way is more promising.
>
> **Other downstream tasks**. Other downstream tasks, e.g., classification and extrapolation, need one embedding of the whole sequence. In this paper, we mainly prove the smoothing effect of CSSC, but not how to encode a sequence into a feature vector, which is non-trivial and is a good future direction. The aforementioned data imputing + sequence encoding could be one way to encode the sequence into one feature and then support other tasks.
>
> **Kernel-based interpolation methods**. We additionally tried two kernel-based interpolation methods: Shukla and Marlin (2019) and Gaussian Process Regression. Their performance is shown in the following table. We find GP lags behind our method in most cases except for a competitive result for toy data at 50% observation. We suspect that as the observation density is high and the trajectory is very smooth, the kernel-based method is already competitive.
> Shukla and Marlin’s method is not quite suitable for our case, because it firstly uses kernels to interpolate the observations into the uniform spaced sequence, then uses RNN to do sequence classification. So we have to modify its codebase to interpolate at the irregular points, however, its result is not good,  which might because its RNN part cannot be used for irregular points interpolation.
>
> | Methods                    | Toy 10%      | Toy 30%      | Toy 50%      | MuJoCo 10%   | MuJoCo 30%   | MuJoCo 50%   | MVMnist 20%  | MVMnist 30%  |
> | --------------------------- | ------------ | ------------ | ------------ | ------------ | ------------ | ------------ | ------------ | ------------ |
> | Shukla and Marlin           | 0.546853     | 0.524762     | 0.499526     | 0.044147     | 0.042352     | 0.042041     | 0.036362     | 0.033716     |
> | Gaussian Process Regression | 0.247505     | 0.003840     | **0.000037** | 0.022794     | 0.003457     | 0.001258     | 0.319261     | 0.294342     |
> | CSSC                        | **0.024656** | **0.000457** | 0.000128     | 0.006097     | **0.000375** | **0.000087** | -            | -            |
> | Hidden CSSC                 | 0.027073     | 0.001503     | 0.000244     | **0.004554** | 0.000378     | 0.000106     | **0.019475** | **0.015662** |

---

### Official Review · AnonReviewer2 · 2020-10-30
**Interesting paper but technique has strong limitations while writing could be improved**

**Rating:** 5
**Confidence:** 3

**Review:**

The paper proposes to use cubic spline interpolation to smooth out discontinuities outputs and state inferred by ODE-RNNs.

While the approach is interesting, it is ill adapted to noisy data (which is a very strong shortcoming IMO) while the experiments are limited to data sets corresponding to applications with limited impact due to their artificial nature.

The paper has multiple issues which I believe require serious rewriting.

1) The paper has grammatical mistakes including in the abstract and many run-on sentences. The readers could use higher quality writing.

2) The premise that the jumps inside the ODE-RNN are shortcomings or inconsistencies is erroneous. It is expected that their would be jumps with each new observation as the filtration with respect to which the latent state is conditioned changes then with a discontinuity. It would in fact be unexpected to have a continuous posterior. This is for instance what typically happens to stock market stock series any time an earnings report is issued.

3) The experimental section is terse and could really use more diverse datasets. Also comparisons with respect to Gaussian and Neural processes would be welcome. A comparison with a simpler smoothing method such as a convolutions, Fourier and wavelet denoising is also warranted.

---

> ### Author Response · Authors · 2020-11-25
> **Response to R2**
>
> Thanks for R2's valuable advice.
>
> **Ill-adapted to noisy data**. The reason why our standard CSSC is ill-adapted to noisy data is that we set our interpolation curve passing through the observation points. However, in the hidden CSSC, there is no such constraint, so the hidden CSSC has the potential to ignore the noise and be applied to noisy data. The careful investigation for noisy data is our future step. In this work, we focus on the theoretical construction, computation, and error analysis of CSSC.
>
> **Writing**. For the writing, we have corrected the grammatical mistakes as many as we could in the revision.
>
> **Premise**: We do not think our premise is erroneous.
> - Firstly, ODE-RNN's latent state will jump at the new observation because it is a posterior of only the new observation point (due to the markovian assumption, it is not conditioned on more previous observations). However, our smoothing is a posterior of all of the observation points without markovian assumption.
> - Secondly, whether we need to do smoothing depends on our prior knowledge of whether the time series is low-order continuous (or more loosely, low-frequency sequene). If yes, the CSSC serves as one kind of smoothing techniques to get more accurate interpolation. The stock series is high-frequent data when an earning report is issued, so it should not use smoothing. However, if the market is stable at a certain period, we can still use CSSC for that period. More suitable cases for smoothing could be the body temperature and the density of air pollutants in a diffusion process.
> - Finally, as R5 mentioned,  Kidger et al., NeurIPS 2020 "Neural Controlled Differential Equations for Irregular Time Series" use cubic splines on the input data to allow Neural ODE to query data at any time point and produce a smooth hidden state trajectory.
> This method also constructs the smoothed hidden state based on all the observations in the cubic spline step.
>
> Therefore, we think our premise is reasonable.
>
> **More experiments**: We have conducted two kernel-based comparison methods: Gaussian process, and Shukla and Marlin (2019), please refer to the table in response to R4.

---

### Official Review · AnonReviewer5 · 2020-11-09
**Resolving the discontinuity of ODE-RNN outputs using splines**

**Rating:** 7
**Confidence:** 5

**Review:**

The paper builds on ODE-RNN model that allows to represent a time series as a continuous trajectory. The authors address the limitation of the ODE-RNN model that the trajectory is continuous everywhere except the observation points. They introduce a compensation term based on cubic splines that transforms the output trajectory into a continuous one. This approach is also applied to correct a hidden state trajectory. The authors demonstrate better interpolation properties on sparse time series data.

Pros:
+ The paper targets the main drawback of ODE-RNN that both the output and the hidden state have a "jump" at the observation points. The idea of correcting the trajectory or hidden state using cubic splines to make it continuous is novel and interesting.

+ The authors provided the theoretical justification for their method and an interpolation error bound. The authors provided the thorough analysis of the model, including the effect of the numerical differentiation.

+ The authors highlighted the two cases of input data (with the continuity on the input level and on the semantic level) and demonstrated the applicability of CSSC and hidden CSSC in each case.

+ In hidden CSSC, the trajectory is being smoothed out without relying on ground-truth data.

Cons:
- I find the decision to set the outputs exactly to x_k  in standard CSSC to be questionable. Instead of correcting the model predictions, like in hidden CSSC, the model explicitly sets the the predictions to the correct value. First, as authors mention, it restricts the model to the time series with no noise, which has a limited use in practice. Second, with incorrect alpha, the computational capacity is of ODE-RNN is not utilized, and the model reverts to a cubic spline to fit the data. It is visible on figure 5 (alpha <= 10) -- the corrected output \hat(o)(t) fits the trajectory well despite the poor reconstruction by ODE-RNN.
Is it possible to do the interpolation in the input space by setting epsilon_k+ = 0; epsilon_k- = o(t_k+) - o(t_k-), similarly to the hidden CSSC? Then the correction c(t) will only play a role of closing the gap between o(t_k+) and o(t_k-). The output of CSSC will remain continuous, but the outputs are no longer restricted to match x_k, improving the expressive power of the model.

- The paper does not provide the error bars for the quantitative results. It is hard to make conclusions about advantages of CSSC and hidden CSSC over other models.

- The proof of interpolation lower bound requires x(t) to have continuous derivatives up to the forth order. It is quite restrictive. I believe that trajectories from Mujoco and moving MNIST already violate this assumption. Can authors hypothesize in which applications we might be able to assume that the inputs belong to C^4?

- Unlike ODE-RNN, the model cannot be used for extrapolation or in the online setting, because the computation of c_k(t) involves the next time point k+1.

Other comments:

-Was the set of subsampled inputs fixed? Was the same set of sampled points used for all the models? In the experiments, only a small fraction of the time points is used as input, and the model performance may change drastically based on the exact set of sampled points.

-I wanted to mentioned that there is a related work by Kidger et al., NeurIPS 2020 "Neural Controlled Differential Equations for Irregular Time Series". They use cubic splines on the input data, which allows Neural ODE to query the data at any time point and produce a smooth hidden state trajectory.  The current work achieves a similar goal without modifying the inputs.

-In Table 1 in the first column (Toy data set with 10% observations) Latent ODE actually has the lowest MSE of 0.013768, while CSSC has 0.024656.

-Section 4.5 "While standard CSSC can always increase the performance upon ODE-RNN". The increase in performance is not guaranteed. I suggest changing the wording to "Standard CSSC has higher performance than ODE-RNN in all our experiments"


Overall:

I vote for weak acceptance of the paper. The idea of adding a compensation term resolves the important issue of the ODE-RNN model and is valuable to the ICLR community. I appreciate the idea of hidden CSSC, where only the predicted hidden states were used to produce a smooth trajectory in the latent space. In standard CSSC, my concern is that the predictions at observation times \hat(t_k) are explicitly set to x_k, making the model inapplicable to noisy data in real-world applications.

I would like to see the error bars for table 1, where the models were trained on the exact same set of subsampled inputs, but with different random seeds. It will help to understand the gap between the cubic splines and CSSC.

============================================

Added after the author response:

The authors adequately addressed my questions and concerns. I appreciate that the authors provided the error bars for their experiments and tried out the idea of modifying epsilon_k, but I wish the authors left more time for the discussion. I think the paper is a valuable contribution to the domain of irregular time series. I am increasing my score to 7.

---

> ### Author Response · Authors · 2020-11-25
> **Response to R5**
>
> Thanks for the insightful review and detailed comments from R5.
>
> **A modified CSSC without using correct data**: Thanks for your great suggestion that borrow the epsilon setting in the hidden space to output space to avoid the use of correct data. This idea is viable, but may not differ much with hidden CSSC. So we have a quick test of it on the toy dataset, shown in the following table. We can see that CSSC w/o correct point can achieve comparable performance as hidden CSSC at 30% and 50%, but will be worse at 10% observation. This may suggest that applying the trick of hidden CSSC to the standard CSSC is less expressive than hidden CSSC.
>
> | Methods                | Toy 10%      | Toy 30%      | Toy 50%      |
> | ---------------------- | ------------ | ------------ | ------------ |
> | CSSC w/o correct point | 0.101303     | 0.001567     | 0.000275     |
> | CSSC                   | **0.024656** | **0.000457** | **0.000128** |
> | hidden CSSC            | 0.027073     | 0.001503     | 0.000244     |
>
>
> **The error bar**. We compute the error bar by training the same subsampled data with different model initialization in 5 times and report the standard deviation of the MSE as the error bar. Due to the time limit, we finished the error bar for CSSC, shown in the following table (cubic splines do not have an error bar, and the original CSSC score is also displayed for reference).
>
> - | Methods                | Toy 10%           | Toy 30%           | Toy 50%           | MuJoCo 10%        | MuJoCo 30%        | MuJoCo 50%        |
>   | ---------------------- | ----------------- | ----------------- | ----------------- | ----------------- | ----------------- | ----------------- |
>   | Spline                 | 0.801249          | 0.003142          | 0.000428          | 0.016417          | 0.000813          | 0.000125          |
>   | CSSC                   |0.024656    | 0.000457      | 0.000128      | 0.006097      | 0.000375      | 0.000087      |
>   | CSSC + error bar      | 0.026608±0.001978 | 0.000401±0.000030 | 0.000153±0.000014 | 0.005705±0.000352 | 0.000353±0.000043 | 0.000067±0.000010 |
>
>
> **Applications we might be able to assume that the inputs belong to $C^4$**. This error bound has the requirement of $C^4$, which is the same requirement for the error bound of cubic spline interpolation. So generally, whenever cubic spline interpolation can be applied, our method can also work. Although sometimes $C^4$ requirement is not meet, in practice, the interpolation performance is still good for the lower-order continuous curve (or loosely, low-frequency data). In the physical world, many slowly changed signals e.g. body temperature, the density of air pollutant in a diffusion process can be assumed $C^4$. Hence the health and climate data is suitable.
>
> **Not an online method**. The online setting can be one future direction. The simplest extension to an online setting is that whenever a new observation arrives, we regard it as the final point of a sequence and apply CSSC to it.
>
> **Is the subsample of time point fixed?**: Yes, the subsampled input is fixed, the same setting as Rubanova et al., NeurIPS2019 “Latent ODEs for Irregularly-Sampled Time Series”.
>
> **Related Work**: Thanks for pointing out. Added to revision.
>
> **Latent ODE**: Yes, in the 10% observation setting, the latent ode is the best. Modified in revision.
>
> **Changing wording**  Thanks for pointing out. Modified.

---

### Decision · Program_Chairs · 2021-01-07
**Final Decision**

**Decision:**

Reject

**Comment:**

This paper introduces a form of cubic smoothing for use with ODE-RNNs, to remove the jump when new observations occur.  I think this paper's motivation is based on a misunderstanding of what the hidden state of an RNN represents.  Specifically, an RNN hidden state is a belief state, not the estimated state of the system.

I think R2 is right that it's correct for a filter to jump when seeing new data.   It's not a matter of whether the phenomenon being modeled is slow-changing or not.  The filtering output is a belief state, which can change instantaneously even if the true state does not.

The important distinction to make is filtering (conditioning only on previous-in-time data) vs smoothing (conditioning on all data).  The smoothing posterior should generally be smooth if the true state changes slowly.

As R4 notes, all of the tasks are based on interpolation, which is not what the ODE-RNN is trying to do, and the proposed method would make the same predictions as a standard ODE-RNN.  Finally, as R4 notes, "The authors do not provide any experimentation on real-world irregularly sampled time series".